

# DLEU2 facilitates bladder cancer progression through miR-103a-2-5p/SOS1 axis

Yinlong Liu[1], Jian Hu[2], Baochun Liao[1], Zhijian Zhu[2], Yong Liu[3] and Qinghua Pan[2]

[1] Department of Abdominal Surgery, Ganzhou Cancer Hospital, Ganzhou, Jiangxi, China
[2] Department of Medical Oncology, Ganzhou Cancer Hospital, Ganzhou, Jiangxi, China
[3] Department of Clinical Pharmacy, Ganzhou Cancer Hospital, Ganzhou, Jiangxi, China

## ABSTRACT

**Background.** Bladder cancer (BC) represents a life-threatening malignancy within the urinary system. Dysregulated long non-coding RNAs (lncRNAs) play pivotal roles in the advancement of BC. LncRNA *deleted in lymphocytic leukemia 2* (DLEU2) is implicated in the development of various cancers. However, its role and regulatory mechanisms in BC remain unclear. This research aimed to explore the expression, biological function, and molecular mechanisms of *DLEU2* In BC progression.

**Methods.** Expression profiles of lncRNAs, microRNAs (miRNAs), and mRNAs in normal and BC tissues were examined by leveraging the raw data sourced from the NCBI GEO database. Reverse transcription quantitative polymerase chain reaction (RT-qPCR) validated expression levels in BC cells. To evaluate the proliferation and migration capabilities of BC cells, assays such as CCK-8, EdU, Transwell, and scratch were carried out. Luciferase reporter assays examined interactions between *DLEU2* and miR-103a-2-5p and between miR-103a-2-5p with *SOS1*. Protein expression of *SOS1* in BC cells was analyzed *via* western blotting.

**Results.** *DLEU2* was markedly increased in BC tissues. Functionally, *DLEU2* overexpression elevated BC cell proliferation and migration, while its knockdown produced the opposite effects. Mechanistically, *DLEU2* acted as a molecular sponge for miR-103a-2-5p, which targeted *SOS1*. miR-103a-2-5p knockdown enhanced proliferation and migration, while co-knockdown of miR-103a-2-5p and *DLEU2* reversed these effects. Overexpression of *SOS1* also promoted proliferation and migration, which were counteracted by miR-103a-2-5p overexpression. Conversely, *SOS1* knockdown inhibited these processes, with miR-103a-2-5p knockdown reversing this inhibition.

**Conclusions.** These findings demonstrate that DLEU2 facilitates BC progression via the miR-103a-2-5p/SOS1 axis. This study reveals a novel regulatory mechanism underlying BC development and highlights DLEU2 as a potential therapeutic target for BC treatment.

Corresponding author
Qinghua Pan, pantsh362@163.com

## INTRODUCTION

Bladder cancer (BC), a malignant tumor within the urinary system, has a notably high mortality rate, posing a significant global threat to human health (*Sung et al., 2021*). BC primarily originates from the malignant transformation of epithelial cells in the bladder (*Heyes et al., 2020*), with urothelial carcinoma accounting for over 90% of cases (*Dobruch & Oszczudłowski, 2021*). Key risk factors for BC include genetic mutations occurring in oncogenes or/with tumor suppressor genes and lifestyle factors like smoking (*Jubber et al., 2023*; *Pullen Jr, 2024*). Despite advancements in surgical techniques and traditional therapies including radiation and chemotherapy, the overall survival rate for BC remains unsatisfactory (*Lenis et al., 2020*). Early diagnosis is critical to improving patient outcomes (*Pandolfo et al., 2024*), highlighting the need to uncover the molecular mechanisms underlying BC progression and to develop novel molecularly targeted therapies.

Lately, research has identified lncRNAs as crucial elements in the onset and progression of various cancers. Many long non-coding RNAs (lncRNAs) are now recognized as potential tumor biomarkers (*Mehmandar-Oskuie et al., 2023*). In BC, lncRNAs such as UCA1 (*Ding et al., 2021*; *Zhen et al., 2017*) and MALAT1 have demonstrated clinical relevance. UCA1 is a potential diagnostic marker for BC while *MALAT1* is overexpressed in BC tissues and correlates with poor overall survival and increased metastatic potential (*Su et al., 2023*; *Xie et al., 2017*). Among oncogenic lncRNAs *deleted in lymphocytic leukemia 2* (DLEU2) is emerging as a key regulator in cancer progression. Evidence suggests that DLEU2 promotes critical tumor traits such as proliferation, migration, invasion, along with resistance to apoptosis in various cancers (*Xu et al., 2021*). Targeting the aberrant expression of DLEU2 offers a promising strategy for early diagnosis and improved patient prognosis (*Qu et al., 2023*). However, the role of *DLEU2* in BC remains largely unexplored.

LncRNAs exhibit diverse and complex mechanisms of action, functioning as molecular decoys, guides, scaffolds, and regulators of signaling pathways to influence various biological processes (*Kopp & Mendell, 2018*). A well-documented mechanism involves lncRNAs acting as competing endogenous RNAs (ceRNAs), where they bind miRNAs to regulate downstream mRNA expression. This lncRNA-miRNA-mRNA interaction axis has been acknowledged as an essential regulatory mechanism in cancer development and progression. For instance, DLEU2/miRNA/mRNA interaction axis is implicated in multiple cancers (*Wu et al., 2024*; *Xiang, Zhang & Li, 2024*). Similarly, the miR-103a-2-5p/SOS1 axis has been shown to regulate cancer progression (*Chen, Yao & Zhou, 2019*; *Yu et al., 2024*). However, whether *DLEU2* serves as a molecular sponge for miR-103a-2-5p to mediate BC progression *via* the *miR-103a-2-5p/SOS1* axis, remains unknown. In this study, we first identified lncRNAs associated with BC through the analysis of the GEO database. We then evaluated the expression with biological function of DLEU2 in BC. Finally, we verified the regulatory mechanism of *DLEU2* in BC progression, focusing on its interaction with the *miR-103a-2-5p/SOS1* axis. This study not only uncovers novel mechanisms underlying BC development, but also provides valuable insights into potential clinical applications for early diagnosis, targeted therapies, and personalized treatment strategies.

## MATERIAL AND METHODS

### Microarray raw data analyses

Employing the raw data sourced from the NCBI GEO database, the expression patterns of lncRNAs, miRNAs, and mRNAs were examined in both normal bladder tissues and BC tissues. Differentially expressed lncRNAs, miRNAs, and mRNAs were detected according to the standards of $P < 0.05$, and |fold change| $\geq 2$. Heat maps, volcano plots, and Venn diagrams were constructed to visualize the results of differential expression analyses. Interactions between lncRNA and miRNA, as well as between miRNAs with mRNAs were predicted using databases such as lncbase, targetscan, miRDB, and BiBiServ.

### Cell culture with transfection

Human BC cells (5637, T24, SW780, J82, and 253J) along with the normal bladder epithelial cell line (SV-HUC-1) were procured from the Shanghai Academy of Life Science (Shanghai, China). Cells were cultured in RPMI 1640 medium (Gibco) fortified with 10% FBS (Gibco) and 1% (penicillin + streptomycin) (Sigma) at 37 °C with 5% $CO_2$.

Small interfering RNAs (SiRNAs) targeting DLEU2 (si-DLEU2-1, si-DLEU2-2, si-DLEU2-3) and SOS1 (si-SOS1) negative control siRNA (si-NC), DLEU2 overexpression (ov-DLEU2), SOS1 overexpression (ov-SOS1), and their corresponding negative control vectors (pcDNA3.1 plasmid) (ov-NC), miR-103a-2-5p mimic, miR-103a-2-5p inhibitor, and miR mimic control (mimic NC), along with inhibitor NC, were procured from GenePharma (Shanghai, China). Transfections were carried out using Lipofectamine 2000 Reagent (Invitrogen) following the manufacturer's protocol. Transfection efficiency was confirmed *via* reverse transcription quantitative polymerase chain reaction (RT-qPCR). All experiments were performed in triplicate. Detailed information on cell lines is available at https://www.cellosaurus.org/. The siRNA sequences are listed in Table 1.

### RT-qPCR

Total cellular RNA was isolated from cells through the utilization of the TRIzol reagent (Invitrogen) with DNase treatment. RNA purity (OD260/OD280 > 1.8) was assessed using a BioPhotometer Plus spectrophotometer (Eppendorf, Hamburg, Germany), and RNA integrity was appraised by subjecting it 1% denatured agarose gel electrophoresis. Reverse transcription was carried out using the MicroRNA RT (Applied Biosystems) for miR-103a-2-5p and the RT Reagent kit (TaKaRa) for lncRNAs and mRNAs. Quantitative PCR (qPCR) was executed with the ABI PCR system (Applied Biosystems). Fold changes in the transcripts were computed using the $2^{-\Delta\Delta CT}$ method. U6 served as the internal reference for miRNA (miR-103a-2-5p; NR_029519.1), and GAPDH for mRNA (DLEU2; NR_152566.1 and SOS1; NM_001382394). Primer sequences are detailed in Table 2 and were validated using BLAST.

### EdU assays

BC cells were transfected for 48 h and co-cultured with EdU reagent (RiboBio) for 2 h. Cells were fixed using 4% formaldehyde, permeabilized, and stained with Apollo 643 reagent. To visualize the nuclei, they were counterstained with DAPI (Sigma), and images of these cells were captured employing a fluorescence microscope (Olympus).

**Table 1  siRNA sequence.**

| Gene | Sequence (5′–3′) |
|---|---|
| si-DLEU2-1 | TGCTGAAACTGCACAAAAATCG |
| si-DLEU2-2 | TAGAGAATAATGGAATGTAAACT |
| si-DLEU2-3 | TGCCTTTTGTTTGCATAGTTTAA |
| si-NC | TTCTCCGAACGTGTCACGTTT |
| si-SOS1 | TCCATTACTTTGAACTTTTGAAG |

## Cell proliferation assays

Cell proliferation assay was assessed by means of the Cell Counting Kit-8 (CCK-8) method. Transfected BC cells were seeded in plates and incubated for 0 and 24 h. At each of these time points, 10 µL of CCK-8 solution was added into each well. Subsequently, the plates were further incubated for an additional 2 h at 37 °C. Optical density (OD) was measured at 450 nm using a microplate reader (Bio-Rad).

## Wound healing assays

After transfection, cells were seeded into 24-well plates and cultured for 24 h to form a confluent monolayer. A sterile pipette tip was used to create a wound in the cell layer. After washing, the cells were cultured in complete media for 48 h and wound closure was monitored and photographed. Measurements were repeated at least three times for statistical analysis.

## Transwell assay

Cell migration was assessed using a Transwell migration chamber (Sigma). Cells were seeded in serum-free media within the upper chamber, while the lower chamber filled with RPMI 1640 medium was supplemented with 20% fetal bovine serum (FBS). After 24 h, cells on the upper membrane surface were removed and cells on the lower surface were then fixed with 4% polyformaldehyde (PFA), stained with 0.1% crystal violet (Beyotime), and counted in five random fields under a microscope.

## Luciferase reporter assay

Luciferase reporter assays were performed using the psi-CHECK2 vector (Promega). Constructs containing wild-type (WT) or mutant (MUT) sequences of DLEU2 and SOS1 3′-UTRs were synthesized by GenePharma. The constructs were co-transfected with miRNA mimic or miRNA inhibitors into 239T cells. After 48 h, luciferase activity was measured using the Dual-Luciferase Assay kit (Promega). Renilla luciferase activity was normalized against firefly luciferase activity.

## Western blotting

Total protein extraction was achieved by using RIPA lysis buffer containingprotease inhibitors (Beyotime). Proteins (30 µg per sample) were separated by sodium dodecyl sulfate polyacrylamide gel electrophoresis (SDS-PAGE) and then transferred onto PVDF membranes supplied by Millipore. Membranes were first blocked and then incubated

**Table 2  Primer sequence.**

| Primer name | Forward (5′–3′) | Reverse (5′–3′) | Size |
|---|---|---|---|
| HCG27 | tctctctcctcttctaccct | ctcccaccccctggcccata | 127 bp |
| LUCAT1 | tatttctgacttggctttct | gcactccagcctgggcgaca | 114 bp |
| LINC-PINT | gagacttggggccagtgact | aaactccaacatgcctgatt | 121 bp |
| Linc01578 | atgaagtagacattggtgga | ttcctcaattccccatagta | 109 bp |
| SNPRN | gagtaccagctggtgtgcca | tgggtactgtgttggggctc | 170 bp |
| NEAT1 | gagggccgggagggctaatc | tgggccccgtcccaggccga | 181 bp |
| TUG1 | tttcttctcgtacgcagaac | ggccgctgccgccgccgcct | 129 bp |
| DLEU2 | atattttgggtttatgtata | ggcatttcttcaaaacacaa | 135 bp |
| CECR9 | cttcatcaacttcattcctg | cttagtcagcctcttggtgt | 152 bp |
| FAM215A | tggttgattaggacttgttg | aggtccctgtcgcagccact | 100 bp |
| GAPDH | ccactggcgtcttcaccacc | tgcaggaggcattgctgatg | 165 bp |
| miR-935 | ACACTCCAGCTGGGggcagtggcgggagcgg | CTCAACTGGTGTCGTGGA | 71 bp |
| miR-103a-2-5p | ACACTCCAGCTGGGagcttctttacagtgct | CTCAACTGGTGTCGTGGA | 71 bp |
| miR-6827-3p | ACACTCCAGCTGGGaccgtctcttctgttcc | CTCAACTGGTGTCGTGGA | 71 bp |
| miR-548ab | ACACTCCAGCTGGGaaaagtaattgtggattt | CTCAACTGGTGTCGTGGA | 71 bp |
| miR-629-3p | ACACTCCAGCTGGGgttctcccaacgtaag | CTCAACTGGTGTCGTGGA | 71 bp |
| miR-6782-3p | ACACTCCAGCTGGGcacctttgtgtccccat | CTCAACTGGTGTCGTGGA | 71 bp |
| miR-1250-3p | ACACTCCAGCTGGGacattttccagcccat | CTCAACTGGTGTCGTGGA | 71 bp |
| miR-6754-3p | ACACTCCAGCTGGGcttcacctgcctctgcc | CTCAACTGGTGTCGTGGA | 71 bp |
| miR-6734-3p | ACACTCCAGCTGGGcccttccctcactcttc | CTCAACTGGTGTCGTGGA | 71 bp |
| RAP1A | aacgacttacaggacctgag | tgcagaagattctaaaaagg | 174 bp |
| SOS1 | gagcccctttgtctccaattc | aaagcagcttcctactagtg | 168 bp |
| RASA2 | tgtgaattttcaaaatgtt | tgagctcttttatacatctc | 149 bp |
| PARK2 | tcacttcaggattctgggag | cattgcccccttcgcaggtg | 164 bp |
| CDC42 | tgggactcaaattgatctca | agaacactccacatacttga | 136 bp |
| NFATC2 | acatggaaaacaagcctctg | tggtgttgcccactatcttc | 147 bp |
| NRAS | acctcagccaagaccagaca | ccatacaaccctgagtccca | 118 bp |
| MAP3K4 | gctcttccatgaagccagag | acccaggaatttgcaccttg | 164 bp |
| PRKCA | gcaagcaaaaaaccaaaacc | aaaggaaagggatcccatga | 161 bp |
| AKT3 | aagacaatgaatgattttga | gccacttcatcctttgcaat | 146 bp |

overnight at 4 °C with primary antibodies: anti-SOS1 antibody (1:1,000, ab140621, Abcam) and anti-glyceraldehyde-3-phosphate dehydrogenase (GAPDH) antibody (1:1,500, ab181602, Abcam). After washing, membranes were incubated with horseradish peroxidase (HRP)-conjugated secondary antibodies (1:1,500, ab205718, Abcam) for 2 h at 25 °C. Protein bands were made visible with the help of an enhanced chemiluminescence detection kit (Beyotime) and imaged with the ChemiDoc™ XRS systems (Bio-Rad).

## Statistical analysis

The data underwent analysis by means of IBM SPSS 22.0 and are presented as means ± standard deviation (SD). To determine statistical significance, the $t$-test was utilized when making comparisons between two groups. For comparisons among multiple groups,

one-way analysis of varience (ANOVA) followed by Tukey's *post-hoc* test was employed. A *P* value of less than 0.05 was regarded as indicating statistical significance.

# RESULTS

## DLEU2 was markedly elevated in BC

To identify abnormally expressed lncRNAs in BC, sequencing data from BC tissues ($n = 4$) along with adjacent normal tissues ($n = 4$) were evaluated using the GEO (GSE138118) (https://www.ncbi.nlm.nih.gov/geo/query/acc.cgi?acc=GSE138118). The top 10 differentially expressed lncRNAs were selected (Figs. 1A and 1B). Among them, DLEU2 showed the highest upregulation (22.48 fold change) and CECR9 the most downregulated (4.55 fold change) in J82 BC cells compared to SV-HUC-1 normal bladder epithelial cells, as confirmed by RT-qPCR (Fig. 1C). Considering these observations, DLEU2 was singled out for more in-depth investigation.

## DLEU2 overexpression could facilitate BC cell proliferation, migration

DLEU2 expression was evaluated across BC cell lines to explore its functional role in BC. DLEU2 expression was highest in J82 cells and lowest in T24 cells (Fig. 2A). J82 and T24 cells were thus chosen for subsequent experiments. DLEU2 overexpression was achieved in T24 cells by (Fig. 2B) transfecting an ov-DLEU2 vector, while DLEU2 knockdown in J82 cells was achieved using si-DLEU2. Si-DLEU2-1, demonstrating the most effective knockdown, was used for all further analyses. Functional assays, including proliferation (Edu), transwell, and scratch assays revealed that DLEU2 overexpression significantly enhanced T24 cell proliferation along with migration, while DLEU2 knockdown in J82 cells suppressed these functions (Figs. 2C–2F).

## DLEU2 sponged miR-103a-2-5p

Given the predominant cytoplasmic localization of DLEU2 in J82 cells (Fig. 3A), its potential interaction with miRNAs was investigated. The lncbase and GEO databases identified 9 miRNAs downregulated in BC (Figs. 3B and 3C). qRT-PCR analysis confirmed that miR-103a-2-5p was the miRNA showing the greatest degree of downregulation within BC cells and was inversely correlated with DLEU2 expression (Figs. 3D and 3E). BiBiServ analysis and dual-luciferase assays validated a direct interaction between DLEU2 and miR-103a-2-5p (Fig. 3F). Additionally, RNA binding protein immunoprecipitation (RIP) assays demonstrated significant enrichment of miR-103a-2-5p with DLEU2, confirming their physical interaction (Fig. 3G). These outcomes implied that DLEU2 functions in the capacity of a sponge for miR-103a-2-5p, which is downregulated in BC cells.

## DLEU2 regulates the proliferation and migration of BC by regulating miR-103a-2-5p

Overexpression of miR-103a-2-5p in J82 cells suppressed proliferation and migration compared to control cells (Figs. 4A–4E). Co-transfection of miR-103a-2-5p mimics and ov-DLEU2 reversed the inhibitory effects of miR-103a-2-5p overexpression on these functions, indicating that DLEU2 counteracts the effects of miR-103a-2-5p.

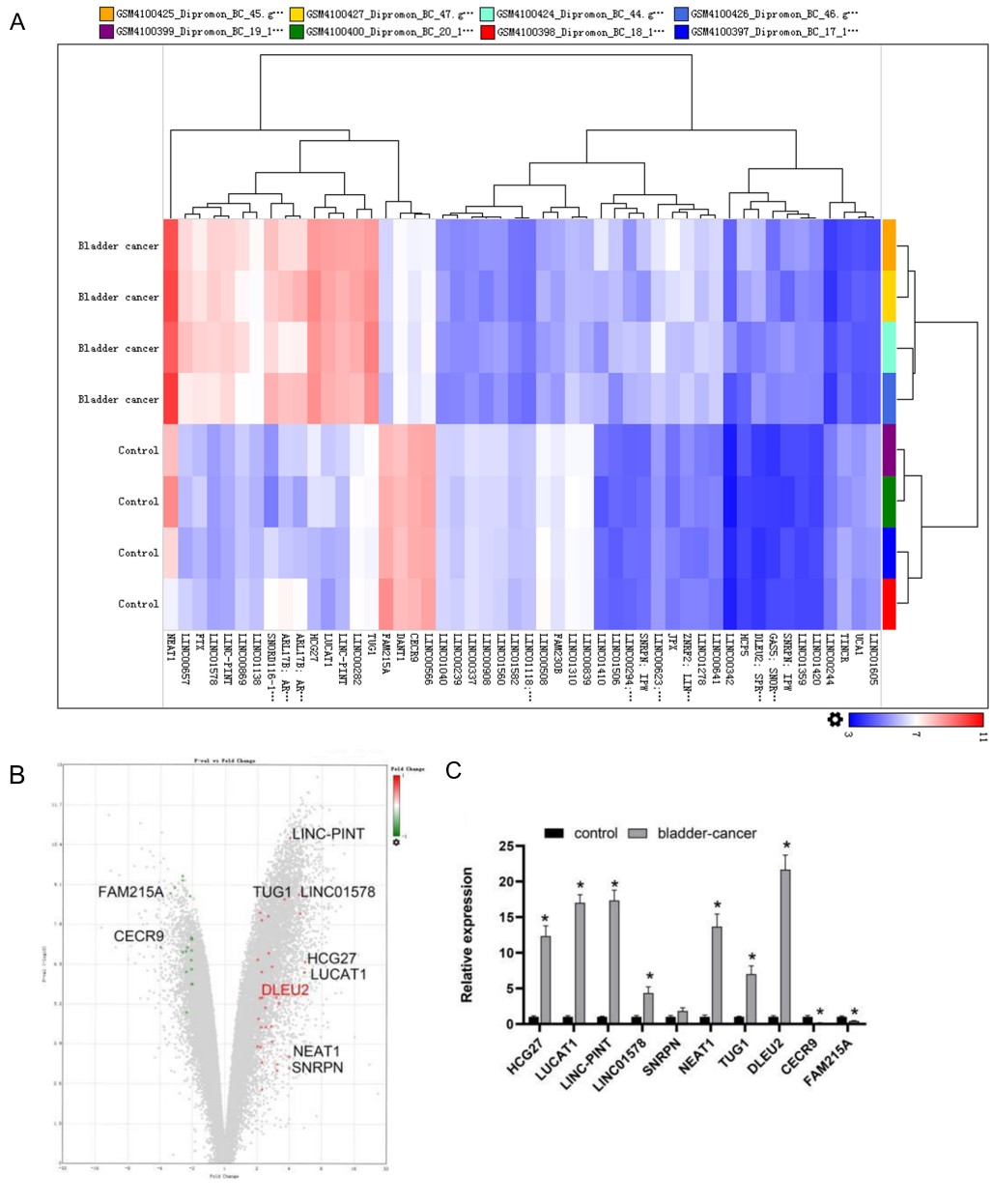

**Figure 1 Bioinformatics analysis and clinical data screening of differentially expressed lncRNAs in the GEO database.** (A) The Top50 lncRNAs differentially expressed in BC and normal tissues were screened from the GEO database. (B) The volcano map showed the GEO analysis data (red was the expression of up-regulated genes, green was the expression of down-regulated genes) and the Top10 differentially expressed genes. (C) The differential expression of TOP10 lncRNAs in SV-HUC-1 cells (control group) and J82 BC cells (bladder-cancer) was assessed by RT-qPCR. Statistical differences were evaluated using $t$-test (*$P < 0.05$).

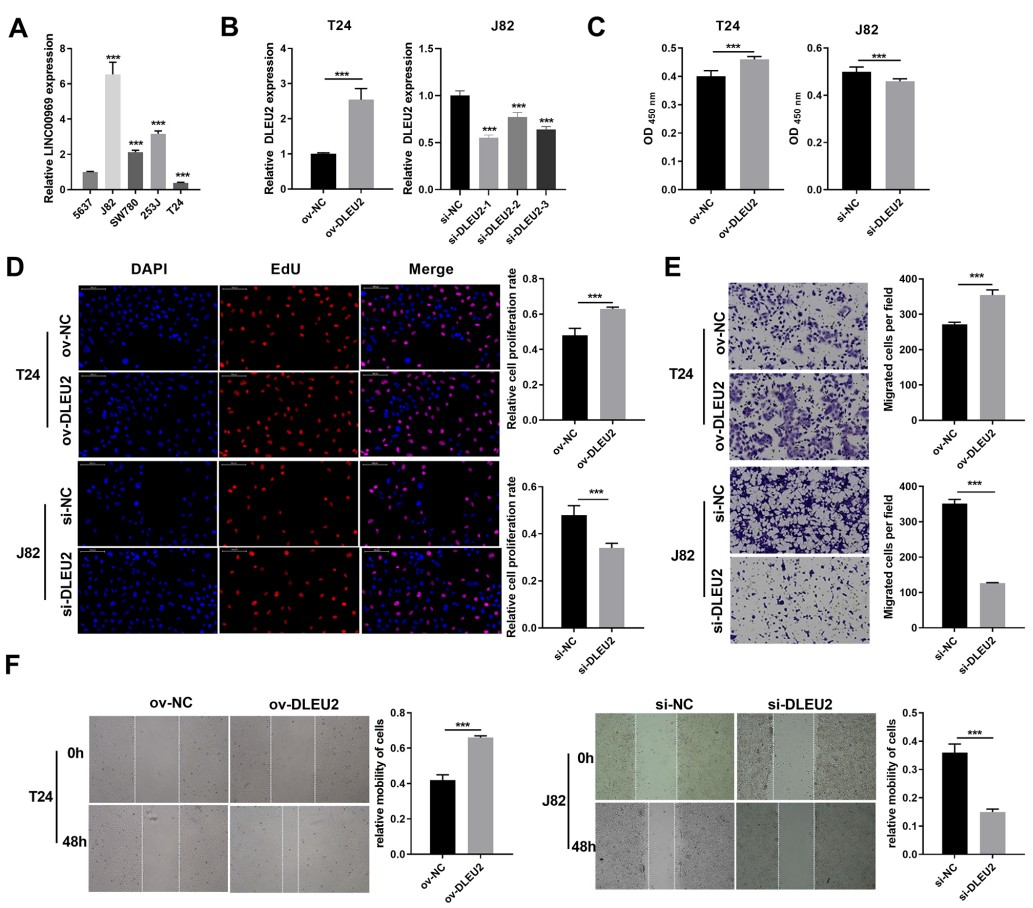

**Figure 2  Effects of DLEU2 overexpression and DLEU2 knockdown on proliferation and migration of BC cells.** (A) The DLEU2 expression in BC cells was measured applying qRT-PCR. Statistical differences were evaluated using one-way ANOVA with *post hoc* analysis was performed by Tukey's multiple comparisons test. (***$P < 0.001$, *vs* 5,637 cell). (B) The DLEU2 expression in BC cells with transforming infection ov-DLEU2 (T24) or si-DLEU2 (J82) was measured applying qRT-PCR. Statistical differences in T24 cells were evaluated using $t$-test. Statistical differences in J82 were evaluated using one-way ANOVA with *post hoc* analysis performed by Tukey's multiple comparisons test. (C and D) Cell proliferation of J82 with T24 cells transforming infection ov-DLEU2 or si-DLEU2 was detected applying CCK-8 or Edu assay. Statistical differences were evaluated using $t$-test. (E and F) Cell migration of J82 with T24 cells transforming infection ov-DLEU2 or si-DLEU2 was detected applying transwell and scratch assay. Statistical differences were evaluated using $t$-test (***$P < 0.001$).

Conversely, miR-103a-2-5p knockdown in T24 cells enhanced proliferation and migration, effects that were mitigated by simultaneous DLEU2 knockdown (Fig. 5). These findings demonstrate that DLEU2 regulates BC cell behavior by modulating miR-103a-2-5p activity.

## SOS1 was a target of miR-103a-2-5p

To identify downstream targets of miR-103a-2-5p, TargetScan software and miRDB were used to predict potential mRNA targets. Integration with GEO data revealed 208 upregulated mRNAs, including 10 associated with the mitogen-activated protein kinases

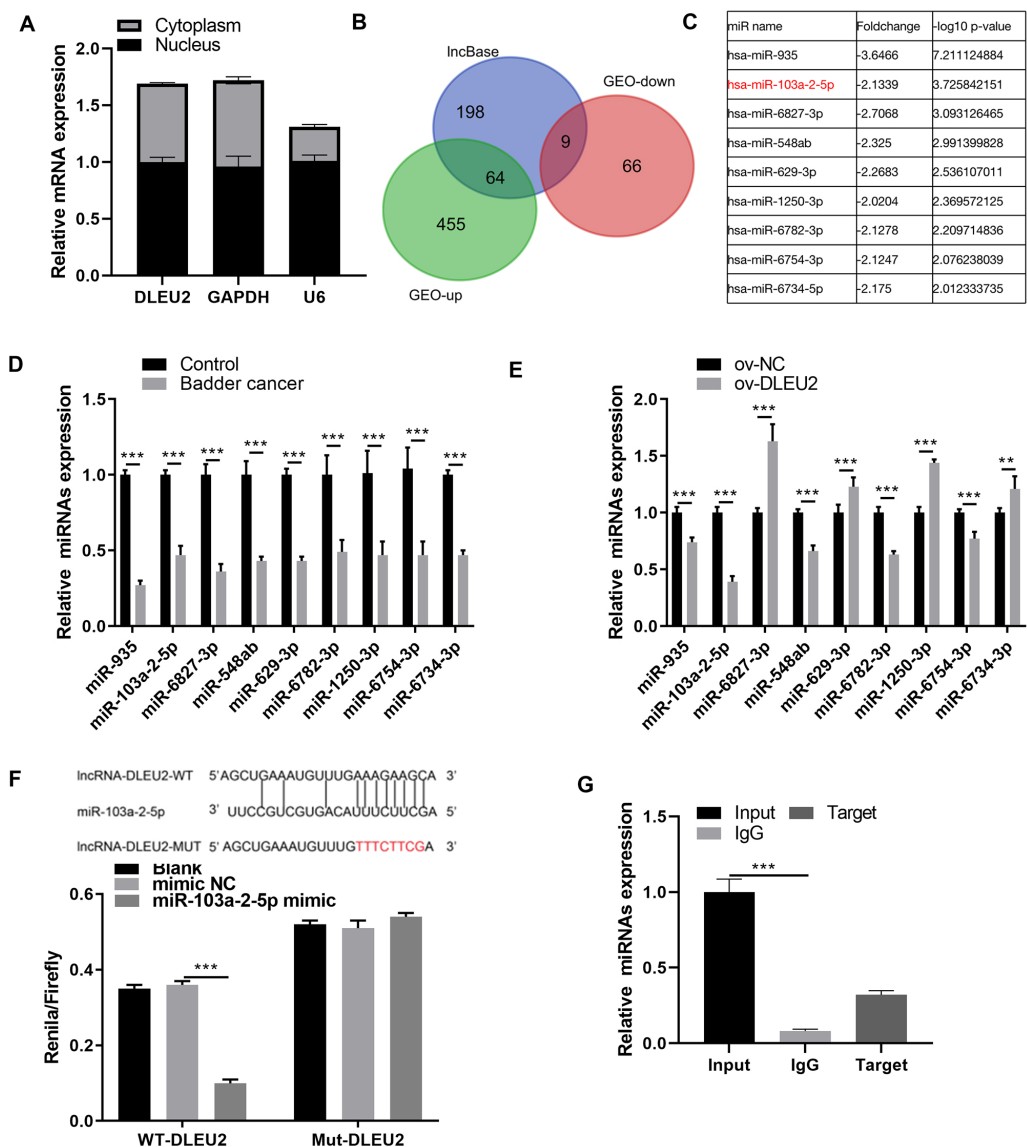

**Figure 3** **Screening miRNAs interacting with DLEU2.** (A) The distribution of DLEU2 in J82 cells was detected by RT-qPCR after karyoplasmic separation. (B) The differentially expressed miRNAs in the GEO database and the miRNAs predicted by lncBase to bind with DLEU2 were intersected to identify a common set of miRNAs. (C) Nine miRNAs were identified by intersecting the significantly downregulated miRNAs in the GEO database with the miRNAs predicted by lncBASE to bind with DLEU2. The fold-change and −log10 $p$-value of mine miRNAs were shown according to GEO database. (D) The nine predicted miRNAs in SV-HUC-1 cells (control group) and T24 BC cells were monitored by qRT-PCR. Statistical differences were evaluated using $t$-test. (E) The nine predicted miRNAs in T24 cell were monitored by qRT-PCR after DLEU2 overexpression. (F) BiBiServ software predicted the binding sites of DLEU2 and miR-103a-2-5p and dual luciferase assay. (G) The binding of DLEU2 and miRNA was detected *via* RIP experiment. Statistical differences were evaluated using $t$-test (**$P < 0.01$ and ***$P < 0.001$).

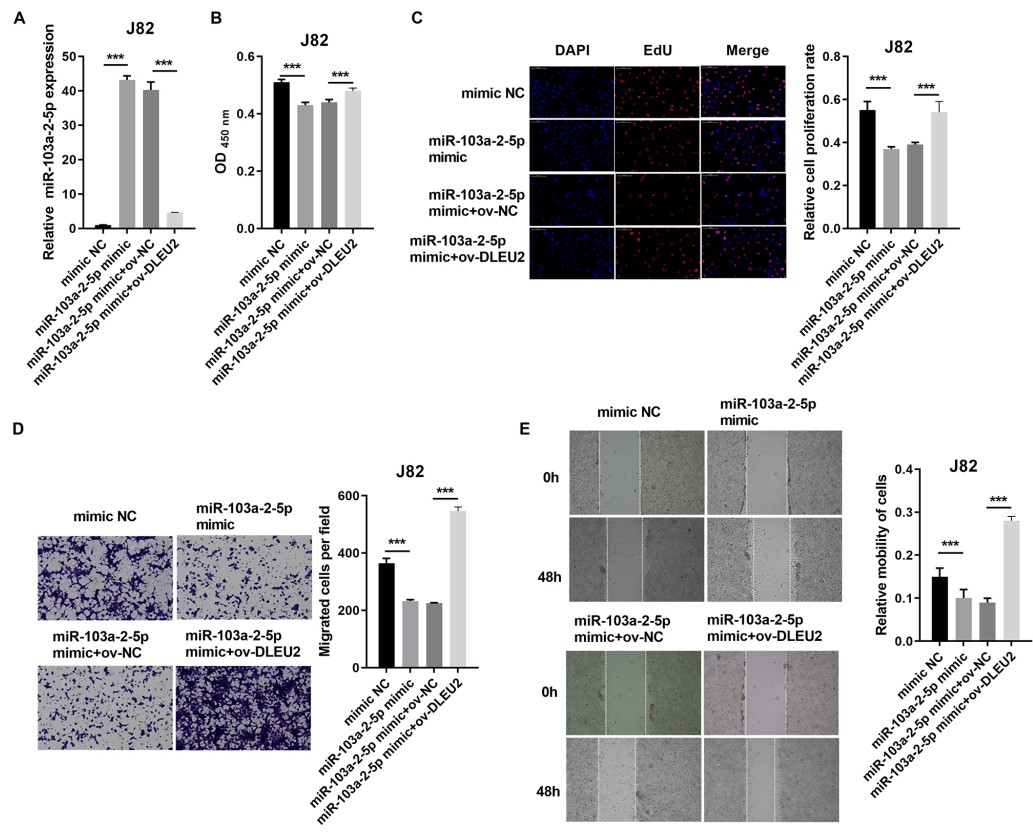

**Figure 4** **DLEU2 overexpression reverses the effect of miR-103a-2-5p overexpression in J82 cells. J82 cell was transformed infection mimic NC, miR-103a-2-5p mimic, miR-103a-2-5p mimic +ov-NC, and miR-103a-2-5p mimic +ov-DLEU2, then used for detection after c.** (A) The miR-103a-2-5p expression in BC cells was measured applying qRT-PCR. (B and C) Cell proliferation of J82 cells was detected applying CCK-8 and Edu assay. (D and E) Cell migration of J82 cells was detected applying transwell and scratch assay. Statistical differences were evaluated using $t$-test (***$P < 0.001$).

(MAPK) pathway, a key pathway altered in BC (Figs. 6A and 6B). Among these, SOS1 exhibited the most significant downregulation upon miR-103a-2-5p overexpression (Fig. 6C). BiBiServ analysis and dual luciferase assays confirmed that miR-103a-2-5p directly binds the SOS1-3′UTR (Fig. 6D). Western blot analysis showed that SOS1 protein levels were highest in J82 cells, while lowest in T24 cells (Fig. 6E); DLEU2 overexpression upregulated SOS1 protein levels in T24 cells, while DLEU2 knockdown reduced SOS1 protein levels in J82 cells (Fig. 6F). miR-103a-2-5p overexpression downregulated SOS1 protein expression, which was reversed by DLEU2 overexpression in J82 cells (Fig. 6G). Conversely, miR-103a-2-5p knockdown upregulated SOS1 protein expression, counteracted by DLEU2 knockdown (Fig. 6G). These results establish SOS1 as a direct target of miR-103a-2-5p and suggest that DLEU2 modulates SOS1 expression through miR-103a-2-5p.

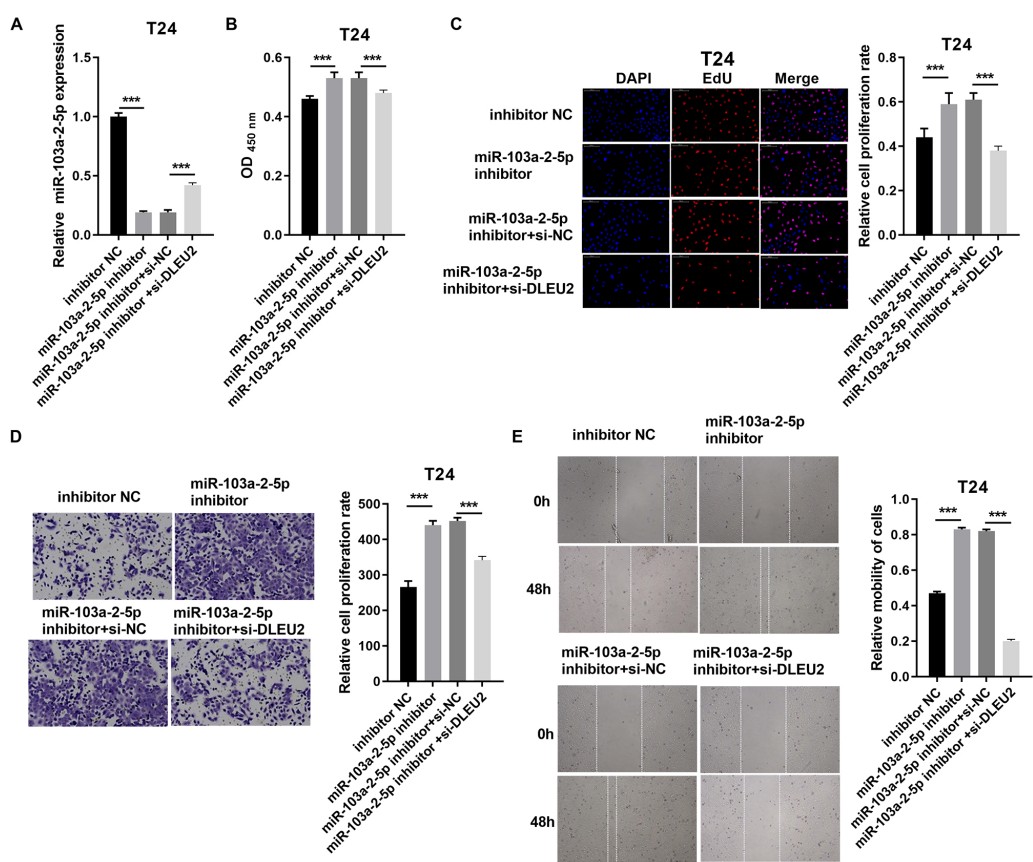

**Figure 5** DLEU2 downexpression reverses the effect of miR-103a-2-5p downexpression in T24 cells. T24 cells were transformed with infection inhibitors NC, miR-103a-2-5p inhibitor, miR-103a-2-5p inhibitor+si-NC, and miR-103a-2-5p inhibitor +si-DLEU2, and then used for d. (A) The miR-103a-2-5p expression in BC cells was measured applying qRT-PCR. (B and C) Cell proliferation of T24 cells was detected applying CCK-8 and Edu assay. (D and E) Cell migration of T24 cells was detected applying transwell and scratch assay. Statistical differences were evaluated using $t$-test (***$P < 0.001$).

## miR-103a-2-5p regulates the proliferation and migration of BC by target SOS1

In J82 cells, SOS1 silencing significantly reduced SOS1 protein expression, as verified by western blot. However, co-treatment with SOS1 knockdown and a miR-103a-2-5p inhibitor restored SOS1 protein levels (Fig. 7A). Functional assay, including cell proliferation, Edu incorporation, transwell migration, and scratch assays demonstrated that SOS1 silencing reduced J82 cell proliferation and migration. Conversely, SOS1 knockdown combined with miR-103a-2-5p inhibition reversed these effects (Figs. 7B–7E). These findings indicate that SOS1 downregulation suppresses J82 cell proliferation and migration, while miR-103a-2-5p inhibition counteracts this suppression.

In T24 cells, SOS1 overexpression increased SOS1 protein expression, whereas co-expression of SOS1 overexpression and miR-103a-2-5p decreased SOS1 protein levels (Fig. 8A). Functional assays revealed that SOS1 overexpression enhanced T24 cells

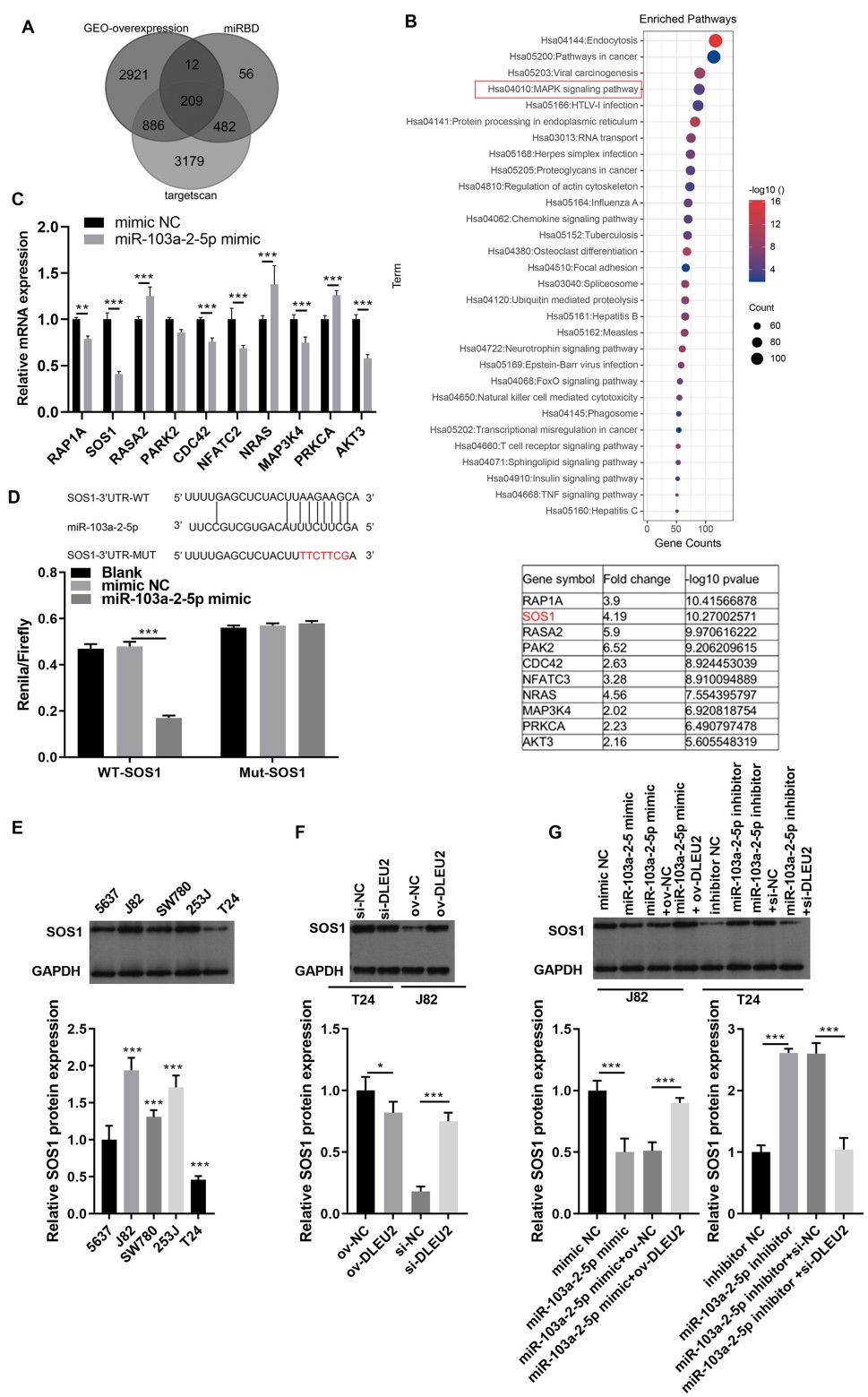

**Figure 6** **SOS1 was a target of miR-103a-2-5p.** (A) Targetscan software, (continued on next page...)

**Figure 6 (…continued)**
miRDB software, and the up-regulated different-expressed mRNA in GEO database simultaneously predicted the target mRNA of miR-103a-2-5p. (B) The map of candidate target genes and MAPK pathway, as well as 10 candidate genes of miR-103a-2-5p. (C) The target effect of miR-103a-2-5p on 10 candidate genes was evaluated by RT-qPCR. Statistical differences were evaluated using $t$-test. (D) The predicted binding sites of SOS1 and miR-103a-2-5p, the targeted relationship between SOS1 and miR-103a-2-5p was monitored by dual-luciferase assay. Statistical differences were evaluated using $t$-test. (E) The SOS1 protein expression in BC cell lines were measured applying Western blot. Statistical differences were evaluated using one-way ANOVA with *post hoc* analysis was performed by Tukey's multiple comparisons test. (F) The SOS1 protein expression in T24 and J82 cells were measured applying Western blot after culture 48 h. T24 cell was transformed infection ov-NC and ov-DLEU2; J28 cell was transformed infection si-NC and si-DLEU2. Statistical differences were evaluated using $t$-test. (G) The SOS1 protein expression in T24 and J82 cells were measured applying Western blot after culture 48 h. J82 cell was transformed infection mimic NC, miR-103a-2-5p mimic, miR-103a-2-5p mimic +ov-NC, and miR-103a-2-5p mimic +ov-DLEU2; T24 cell was transformed infection inhibitor NC, miR-103a-2-5p inhibitor, miR-103a-2-5p inhibitor +si-NC, and miR-103a-2-5p inhibitor +si-DLEU2, then used for detection after culture 48 h. Statistical differences were evaluated using $t$-test (*$P < 0.05$, **$P < 0.01$ and ***$P < 0.001$).

proliferation and migration. However, co-expression of miR-103a-2-5p reversed these effects by suppressing SOS1 expression (Figs. 8B–8E). These outcomes suggest that miR-103a-2-5p acts as a negative regulator of SOS1, thereby modulating BC cell proliferation and migration.

## DISCUSSION

BC is a significant global malignancy of the urinary system (*Lobo et al., 2022*), with rising incidence rates despite advances in treatment, including targeted therapy (*Gill & Perks, 2024*). This underscores the necessity of uncovering the molecular processes responsible for BC progression and identify novel biomarkers and therapeutic targets. Here, we demonstrated that DLEU2, a lncRNA, is markedly upregulated in BC cells, promoting proliferation and migration. Mechanistically, DLEU2 acts as a ceRNA by sponging miR-103a-2-5p, which in turn modulates the expression of SOS1. These findings underscore the latent value of of DLEU2 as a biomarker and therapeutic strategies in BC.

DLEU2 has been extensively studied in multiple cancer types , like gastric cancer, lung cancer, cervical cancer (*He et al., 2021*; *Hu et al., 2022*; *Wu et al., 2020*). It functions as an oncogene, enhancing proliferation, invasion, and chemoresistance in breast and gastric cancers, partly through its interactions with key signaling pathways such as PI3K/AKT (*Hu et al., 2022*; *Islam et al., 2024*). Consistent with these observations, we found that DLEU2 is highly expressed in BC, where it significantly promotes cell proliferation and migration, further reinforcing its role as an oncogene.

LncRNAs are known to function as ceRNAs, sponging miRNAs (*Paraskevopoulou & Hatzigeorgiou, 2016*) to regulate downstream targets. In this study, DLEU2 was identified as a sponge for miR-103a-2-5p, a multifunctional miRNA with context-dependent roles in cancer. For instance, proliferation and migration in esophageal squamous cell carcinoma (ESCC) (*Gao et al., 2020*), but acts as a tumor suppressor in acute myeloid leukemia (AML) and colorectal cancer (CRC) by inhibiting proliferation and inducing apoptosis (*Cen et al., 2024*; *Zhu et al., 2019*). Similarly, in BC, miR-103a-2-5p exhibited tumor-suppressive

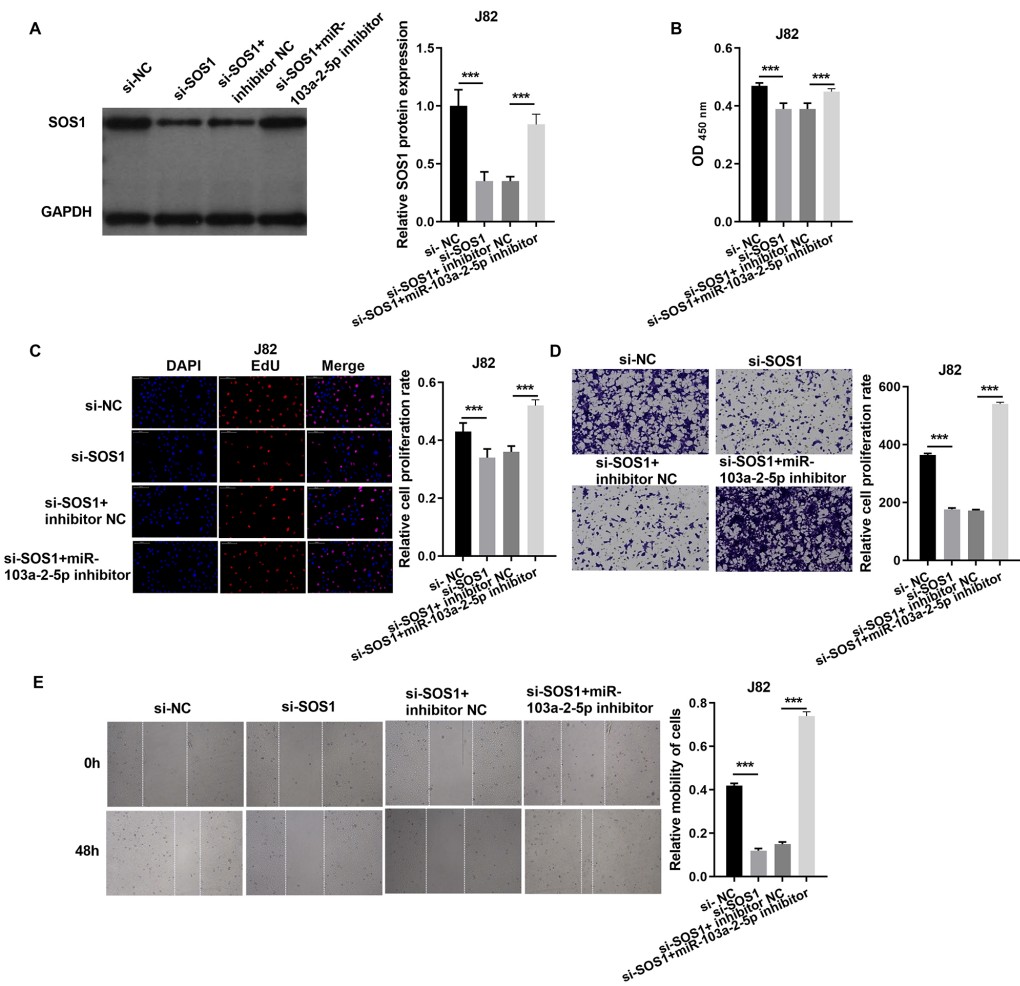

**Figure 7** **SOS1 overexpression reverses the effect of miR-103a-2-5p overexpression in J82 cells.** J82 cells were transfected with si-NC, si-SOS1, si-SOS1 + inhibitor NC, and si-SOS1 + miR-103a-2-5p inhibitor, then used for detection after 48 hours of culture. (A) The SOS1 protein expression in J82 cells were measured applying western blot. (B and C) Cell proliferation of J82 cells were detected applying CCK-8 and Edu assay. (D and E) Cell migration of J82 cells were detected applying transwell and scratch assay. Statistical differences were evaluated using $t$-test (***$P < 0.001$).

activity, as its overexpression inhibited proliferation and migration. Furthermore, DLEU2 overexpression antagonized the effects of miR-103a-2-5p, establishing that DLEU2 promotes BC progression by sponging this miRNA.

MiRNAs regulate target mRNA expression by binding to their 3′ untranslated regions (UTRs), while lncRNAs modulate this process by acting as ceRNAs, preventing miRNA-target interactions. Previous studies have shown that DLEU2 promotes tumorigenesis *via* the miR-212-5p/ELF3 axis and miR-30c-5p/LDHA axes (*Wu et al., 2024*; *Xiang, Zhang & Li, 2024*). Here, we identified SOS1 as a direct target of mir-103a-2-5p. Where the binding of miR-103a-2-5p to SOS1's 3′UTR reduced SOS1 protein levels. DLEU2 sponges miR-103a-2-5p, thereby increasing SOS1 protein levels, which facilitates BC cell proliferation and

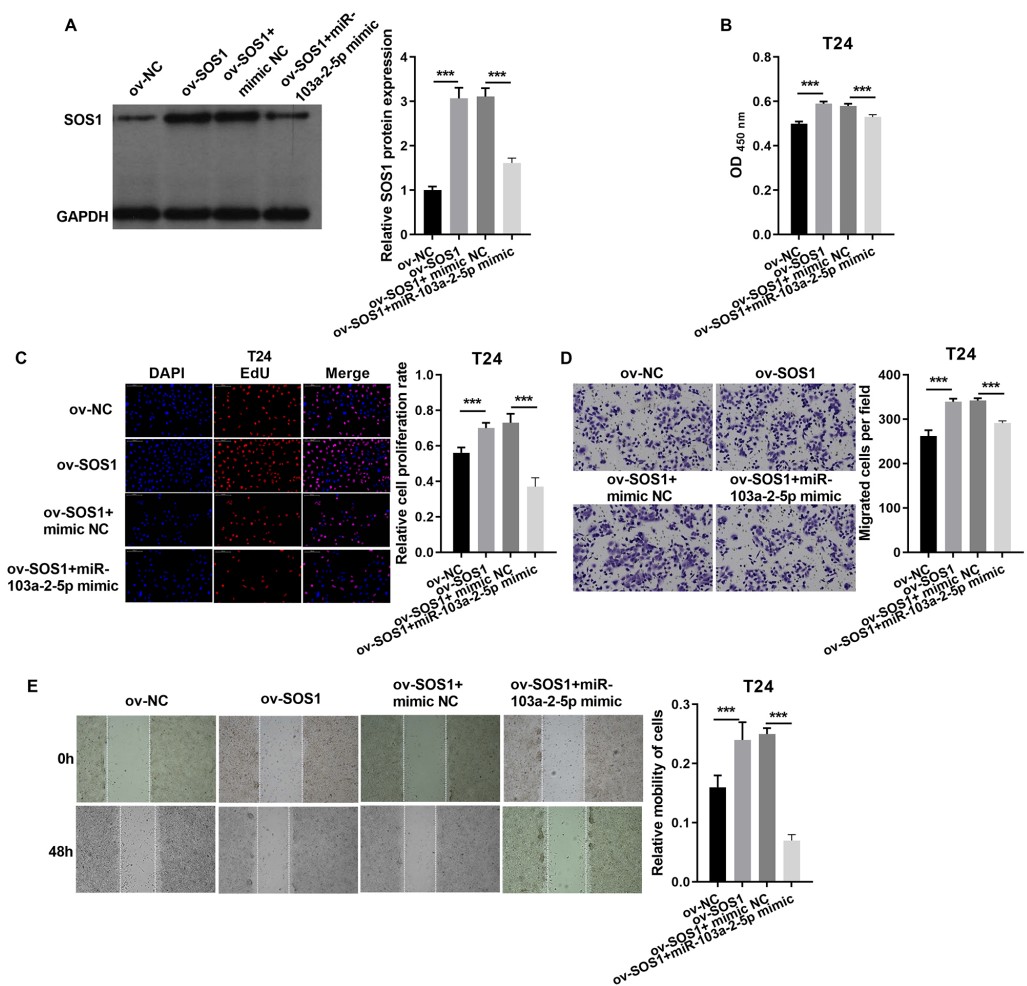

**Figure 8** **SOS1 downexpression reverses the effect of miR-103a-2-5p silence in T24 cells.** T24 cells were transformed/infected with ov-NC, ov-SOS1, ov-SOS1 + mimic NC, and ov-SOS1 + miR-103a-2-5p mimic, then used for detection after 48 hours of culture. (A) The SOS1 protein expression in BC cells were measured applying western blot. (B and C) Cell proliferation of T24 cells were detected applying CCK-8 and Edu assay. (D and E) Cell migration of T24 cells were detected applying transwell and scratch assay. Statistical differences were evaluated using $t$-test (***$P < 0.001$).

migration. This aligns with previous findings implicating SOS1 in the progression of liver, lung, and colorectal cancers (*Daley et al., 2024*; *Li et al., 2022*; *Sudhakar et al., 2024*). SOS1 inhibition has shown promise in delaying cancer progression, highlighting its potential as a therapeutic target. Our results suggest that targeting the DLEU2/miR-103a-2-5p/SOS1 axis may be a viable strategy for BC treatment.

This research has a few drawbacks. The experiments were run on a constrained set of cell lines, and this may impinge upon the ability to generalize the findings. Additionally, the lack of *in vivo* validation precludes a comprehensive understanding of DLEU's role in BC progression. Furthermore, while SOS1 was identified as a key downstream target of DLEU2, other potential mechanisms may exist and warrant further investigation.

Combining DLEU2 with other systemic biomarkers, such as elevated preoperative systemic inflammatory index (SII) levels, may enhance prognostication and guide clinical decision-making for BC patients undergoing radical cystectomy. Future studies should focus on validating these results *in vivo* and exploring additional pathways downstream of DLEU2 to refine therapeutic strategies for BC.

## CONCLUSION

In summary, our study demonstrates that DLEU2 is markedly upregulated in BC, and promotes cell proliferation along with migration *via* the miR-103a-2-5p/SOS1 axis. These findings uncover a novel mechanism underlying BC progression and identify DLEU2 as a potential therapeutic target. This research provides valuable insights into the molecular basis of BC and offers promising directions for early diagnosis, targeted therapy, and personalized treatment strategies.

### Funding

The authors received no funding for this work.

### Competing Interests

The authors declare there are no competing interests.

### Author Contributions

- Yinlong Liu conceived and designed the experiments, performed the experiments, analyzed the data, prepared figures and/or tables, authored or reviewed drafts of the article, and approved the final draft.
- Jian Hu performed the experiments, analyzed the data, prepared figures and/or tables, and approved the final draft.
- Baochun Liao performed the experiments, analyzed the data, prepared figures and/or tables, and approved the final draft.
- Zhijian Zhu performed the experiments, analyzed the data, prepared figures and/or tables, and approved the final draft.
- Yong Liu performed the experiments, analyzed the data, prepared figures and/or tables, and approved the final draft.
- Qinghua Pan conceived and designed the experiments, authored or reviewed drafts of the article, and approved the final draft.

### Data Availability

The data is available at figshare: Liu, Yinlong (2024). DLEU2 facilitates bladder cancer progression through miR-103a-2-5p/SOS1 axis. figshare. Dataset. https://doi.org/10.6084/m9.figshare.27288624.v2.

## Supplemental Information

Supplemental information for this article can be found online at http://dx.doi.org/10.7717/peerj.18995#supplemental-information.

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
