# Peer review of "DLEU2 facilitates bladder cancer progression through miR-103a-2-5p/SOS1 axis"

_PeerJ, doi:10.7717/peerj.18995_

## Round 0.1 · original submission · Major Revisions

Having carefully considered your manuscript and the comments from three reviewers, I believe your work has merit but requires substantial revision before it can be considered for publication. Please carefully address all the reviewers' comments. A point-by-point response to all comments should be included with your revised manuscript.

Reviewer 1 ·

Basic reporting

The study highlights the DLEU2/miR-103a-2-5p/SOS1 axis as a novel pathway in bladder cancer progression. Some suggestions have been provided:

* The manuscript contains grammatical and typographical errors that hinder clarity.
* Sentences in the introduction and discussion lack flow and readability, requiring professional editing
* The introduction should be shortened. Leave for discussion all the reported details. In addition, while some examples are mentioned (e.g., MALAT1, UCA1), the context of how this study fills a knowledge gap should be emphasized.

Experimental design

The investigation is rigorous and adheres to high technical standards. However, the manuscript would benefit from more detailed descriptions of certain methodological steps, particularly the criteria for cell line selection and the statistical analysis workflow, to ensure full replicability. Ethical standards are sufficiently addressed.

Validity of the findings

The article provides robust data and statistically sound analyses; however, it lacks a clear discussion of the broader impact and novelty of its findings within the current literature. Including more context on how the DLEU2/miR-103a-2-5p/SOS1 axis advances knowledge in bladder cancer would enhance its value. Additionally, meaningful replication studies should be encouraged, with a clear rationale for their benefit to the field.

Additional comments

* A conclusion sections should be listed separately
* In addition to molecular insights like the DLEU2/miR-103a-2-5p/SOS1 axis, systemic biomarkers such as the SII have shown prognostic significance in bladder cancer. The study by doi: 10.3390/medicina59122063. demonstrated that elevated preoperative SII levels were associated with worse recurrence-free and overall survival outcomes in patients undergoing radical cystectomy, reinforcing the utility of integrating molecular and systemic biomarkers to improve prognostication and guide clinical decision-making
* Figures 1-8 could provide more context about experimental conditions
* Clearly describe statistical methods and ensure p-values are appropriately reported in all figures
* Cite the doi: 10.3390/cancers16061115. article to emphasize the importance of developing standardized diagnostic and therapeutic approaches across all urological malignancies, including bladder cancer. The variability highlighted in guidelines parallels the need for more robust biomarker-based frameworks like the one proposed in this study.

Reviewer 2 ·

Basic reporting

Please see Additional comments.

Experimental design

Please see Additional comments.

Validity of the findings

Please see Additional comments.

Additional comments

This manuscript describes an interesting finding: the DLEU2 facilitates bladder cancer progression through miR-103a-2-5p/SOS1 axis. The experiments are generally well-designed. However, some issues need to be addressed.
1.Abstract: Why choose DLEU2 as the research object? The clear, sharp objective is missed in the Abstract.
2.Introduction: The research background of DLEU2 in tumors should be described in more detail. The research background of miR-103a-25-p/SOS1 in tumors should be briefly described. The regulatory mechanism of DLEU2 in tumors should be briefly introduced. The research purpose and clinical significance of this project should be briefly described.
3.Methods: What statistics are used for pairwise analysis between multiple groups?
4.Results: In “DLEU2 overexpression could facilitate BC cells proliferation, migration”, the description of cells in the article is different from the figure 2.
5.Results: In “miR-103a-2-5p inhibitors were transfected in J82 cells, and the RT-qPCR data declared miR-103a-2-5p downexpression in T24 cells (Fig. 5A)”, in this paragraph, J82 cells should be T24 cells.
6.Results: “Moreover, the western blot experiment results confirmed that SOS1 protein was also highly expressed in BC cell lines J82 and T24 (Fig. 6E)”, please check if the result is correct.
7.Results: “DLEU2 overexpression could increase SOS1 protein expression, while DLEU2 knockdown could decrease SOS1 protein expression”, are overexpression and interference occurring in the same cell?
8.Results: Please redescribe the results of Figure 6G.
9.There are two Figure 5.
10.Results: Please redescribe the results of Figure7A.
11.Discussion: Focus on comparing the differences between the role and regulatory mechanism of DLEU2 in different tumors and in this study?
12.Discussion: Focus on comparing the differences between the role and regulatory mechanism of miR-103a-2-5p/SOS1 in different tumors and in this study?
13.Discussion: limitations?
14.The conclusion only reaffirms the research findings, but does not address broader implications or provide specific recommendations for future study.
15.Grammatical and spelling errors need revised.

Reviewer 3 ·

Basic reporting

1.1 This study investigated the role of long non-coding RNA DLEU2 in bladder cancer (BC) progression. Although the overall logic of the manuscript is clear, it is still necessary to improve the level of English writing so that the reader can better understand the content.
1.2 The references are reasonable and the research background is basically valid, but the background of the abstract can be more focused on the specific role of DLEU2 in BC. It would be helpful to mention any existing research suggesting a potential role for dule2 in BC or other cancers. Therefore, the introduction is somewhat scattered in its focus, jumping between different aspects of BC research and lncRNA roles. It could benefit from a clearer focus on the specific research question and the rationale for investigating DLEU2's role in BC.
1.3 The structure of the manuscript is complete, and it is recommended to separate the conclusions into one chapter. In addition, The resolution of the images in Figures 1 and 6 should be improved. The legend for all diagrams needs to be more detailed.Raw data is available.

Experimental design

The study found that DLEU2 is upregulated in BC cells and promotes cell proliferation and migration. Further investigation revealed that DLEU2 acts as a sponge for microRNA miR-103a-2-5p, which in turn regulates the expression of SOS1, an oncogene implicated in BC. The study concludes that DLEU2 facilitates BC progression through the miR-103a-2-5p/SOS1 axis, suggesting a potential therapeutic target for BC treatment.
2.1. Providing specific details about the GEO datasets used, including accession numbers, is essential for transparency and reproducibility. Providing primer sequences for DLEU2, miR-103a-2-5p, and SOS1, along with BLAST validation information and accession numbers, is crucial for reproducibility. Clearly describe how the data was normalized, including the housekeeping gene used (e.g., GAPDH, β-actin) and its rationale (e.g., stability across different conditions).
2.2. It's essential to provide detailed information about the cell lines used, including their origin (e.g., tissue source), passage number, and any specific characteristics relevant to the study (e.g., known mutations, sensitivity to certain treatments). This helps readers assess the relevance of the findings to other cell lines and potentially translate them to clinical scenarios.
2.3. Elaborating on transfection methods (e.g., lipid-based, electroporation) and the specific siRNAs used (e.g., sequence, target site) is important. Also, mention the methods used to determine transfection efficiency (e.g., fluorescent microscopy, qPCR) and the percentage of transfected cells in the experiments.
2.4 The discussion doesn't delve into the specific molecular mechanisms by which DLEU2, miR-103a-2-5p, and SOS1 interact. For example, it could explain how the interaction between DLEU2 and miR-103a-2-5p leads to changes in SOS1 expression and how these changes impact BC cell behavior.
2.5 The discussion should acknowledge the limitations of the study, such as the use of a limited number of cell lines, the lack of in vivo validation, and the potential for confounding factors.

Validity of the findings

3.1 The results in Figure 1 indicated that “the RT-qPCR data showed that the DLEU2 was most differentially high expressed within J82 BC cells versus to that within SV-HUC-1 cells”. The description of this sentence is incorrect because the expression of some LncRNAs is reduced in BC cells.
3.2 There appears to be an inconsistency between the results presented in Figure 2 and the text. Additionally, the authors should clarify the expression levels of DLEU2 in SV-HUC-1 and T24 cells. If DLEU2 expression is indeed higher in T24 cells compared to SV-HUC-1 cells, then the functional studies could be streamlined by focusing solely on DLEU2 knockdown in T24 and J82 cells. Overexpression experiments in additional cell lines might not be necessary."
3.3 While mentioning the use of qRT-PCR to detect the nine predicted miRNAs after DLEU2 overexpression in Figure 3D, the study doesn't show the expression levels of these miRNAs in control cells, making it difficult to assess the significance of the observed changes. Performing RNA immunoprecipitation (RIP) experiments, which would allow for the physical isolation and detection of DLEU2-miR-103a-2-5p complexes, would provide stronger evidence for their direct interaction.
3.4 The author indicated that “The results of GEO RNA sequencing in BC showed significant changes in the MAPK pathway. Therefore, the intersection of 208 candidate target genes and the MAPK pathway was screened, and the results showed that 10 genes in the MAPK pathway might be the target genes of miR-103a-2-5p (Fig.6B).” “ and the results showed that the significant target of miR-103a-2-5p was SOS1(Fig. 6C).”. The descriptions are not sufficiently detailed, leading to confusion. Authors should provide a more detailed explanation and interpretation of these research findings. In addition, the authors should provide further evidence to support the validity of this conclusion “Most importantly, overexpression or silence of miR-103a-2-5p could negatively regulate SOS1 protein expression and positively regulate by DLEU2 expression (Fig. 6G).”
3.5 The discussion relies on general statements about lncRNAs and their roles in cancer, such as "multifarious lncRNAs are continually aberrantly expressed in BC pathological development, plays critical role in gene regulation." While true, these statements lack specificity and don't provide a clear connection to the study's findings.

Additional comments

Abstracts would benefit from more refined and accurate descriptions.such as: the background should clearly explain the rationale behind selecting DLEU2 for their study. In addition, the conclusion is a bit too brief and could benefit from providing more specific details, discussing broader implications, and suggesting future research directions.

---

## Round 0.2 · Minor Revisions

We have reviewed your revised manuscript along with the reviewers' evaluations. While all three reviewers recommend acceptance, upon careful assessment, two minor issues still need to be addressed:
1. The manuscript requires professional English language editing to enhance clarity and readability
2. Please add the Figshare data citation in appropriate sections of the manuscript
Please provide these final revisions. Upon receiving the updated version addressing these points, we will proceed with the formal acceptance of your manuscript.

Reviewer 1 ·

Basic reporting

Clear and professional English used throughout.
Relevant literature references and sufficient background context are provided.
The structure is in line with PeerJ standards, and figures/tables are well-labeled and of good quality.
Raw data is included and appropriately referenced.

No comments. The reporting is robust.

Experimental design

good work, no comment

Validity of the findings

The findings are valid, supported by robust data, and clearly linked to the research question. No comment.

Additional comments

none

Reviewer 2 ·

Basic reporting

Revisions to the manuscript are acceptable without further comment.

Experimental design

no comment

Validity of the findings

no comment

Reviewer 3 ·

Basic reporting

The author's revision is generally satisfactory.

Experimental design

The author's revision is generally satisfactory.

Validity of the findings

The author's revision is generally satisfactory.

---

## Round 0.3 · accepted · Accept

Thank you for your thorough revision. The manuscript has adequately addressed all previous concerns and significantly improved in clarity and scientific rigor. We are pleased to accept this paper for publication.